# Trimetazidine, an Anti-Ischemic Drug, Reduces the Antielectroshock Effects of Certain First-Generation Antiepileptic Drugs

**DOI:** 10.3390/ijms231911328

**Published:** 2022-09-26

**Authors:** Kinga Borowicz-Reutt, Monika Banach

**Affiliations:** Independent Unit of Experimental Neuropathophysiology, Department of Toxicology, Medical University of Lublin, PL 20-090 Lublin, Poland

**Keywords:** trimetazidine, first-generation antiepileptic drugs, maximal electroshock, drug interactions

## Abstract

Trimetazidine (TMZ), an anti-ischemic drug for improving cellular metabolism, is mostly administered to patients with poorly controlled ischemic heart disease (IHD). Since IHD is considered the most frequent causative factor of cardiac arrhythmias, and these often coexist with seizure disorders, we decided to investigate the effect of TMZ in the electroconvulsive threshold test (ECT) and its influence on the action of four first-generation antiepileptic drugs in the maximal electroshock test (MES) in mice. The TMZ (up to 120 mg/kg) did not affect the ECT, but applied at doses of 20–120 mg/kg it decreased the antielectroshock action of phenobarbital. The TMZ (50–120 mg/kg) reduced the effect of phenytoin, and, when administered at a dose of 120 mg/kg, it diminished the action of carbamazepine. All of these revealed interactions seem to be pharmacodynamic, since the TMZ did not affect the brain levels of antiepileptic drugs. Furthermore, the combination of TMZ with valproate (but not with other antiepileptic drugs) significantly impaired motor coordination, evaluated using the chimney test. Long-term memory, assessed with a passive-avoidance task, was not affected by either the TMZ or its combinations with antiepileptic drugs. The obtained results suggest that TMZ may not be beneficial as an add-on therapy in patients with IHD and epilepsy.

## 1. Introduction

Trimetazidine (1[2,3,4-trimethoxy-benzyl] piperazine, 2 HCl; TMZ) is a widely known and commonly used cytoprotective, anti-ischemic, and anti-anginal drug recommended as a prophylactic and add-on therapy for the symptomatic treatment of patients with stable angina pectoris, chorioretinal disturbances, and vertigo [1]. The effects of TMZ are primarily based on its antioxidant action. This drug was shown to improve mitochondrial functions by reducing excessive release of ROS and inorganic phosphates [2]. Even single applications of TMZ (5 and 25 mg/kg) significantly increased superoxide dismutase (SOD) activity and ROS scavenging in rat brains [3]. The cytoprotective properties of TMZ can be attributed to the improvement of mitochondrial respiration that it produces. Specifically, TMZ selectively inhibits long-chain 3-ketoacyl-co-enzyme A (CoA) thiolase (3-KAT), which catalyzes the terminal step of fatty acid β-oxidation. This shifts cardiac metabolism from fatty acid to glucose oxidation. It should be stressed that free fatty acid metabolism increases oxygen consumption by 11–12%, which overwhelms the oxidative capacity of the heart [4].

TMZ has also been reported to prevent calcium overload and the development of intracellular acidosis in cardiomyocytes. The former effect is related to decreased binding of calcium ions to the mitochondrial permeability transition pores, which reduces excessive ion flux and damage to the mitochondrial membrane [1,3,5]. It is likely because of these mechanisms that TMZ was observed to induce axonal myelination and regeneration in a rat model of Alzheimer’s disease. In the same study, TMZ increased the expression of the DHCR24 (seladin-1) gene, an accepted marker of oxidative stress and degeneration in the brain [6]. In general, TMZ is very well tolerated; however, due to the blockade of D_2_ receptors, it can induce extrapyramidal motor disturbances, gait disorders, and tremors. Therefore, this drug is contraindicated in patients with Parkinson’s disease and Parkinsonian symptoms [7]. Under experimental conditions, TMZ has demonstrated gastroprotective, hepatoprotective, anti-inflammatory, antinociceptive, and antiapoptotic effects [8,9]. Interestingly, this anti-ischemic drug also affected glucose metabolism in rat brains, significantly increasing [^3^H]glucose uptake in the striatum, hippocampus, frontal cortex, thalamus, hypothalamus, pons, medulla oblongata, and cerebellum [10]. Hence, central effects resulting from treatment with TMZ cannot be excluded.

A growing body of evidence indicates that oxidative stress and mitochondrial dysfunctions may be either consequences or causative factors of seizures [11]. In a study by Sedky et al. [2], eight-week oral treatment with TMZ (20 mg/kg per day) provided an anticonvulsant effect against pentylenetetrazole (PTZ)-kindled seizures in rats and potentiated the action of valproate in this model. Moreover, this anti-ischemic drug decreased insulin resistance and hyperlipidemia induced in rats by seizures and valproate treatment [2]. In mice, TMZ (10 and 20 mg/kg) applied orally before PTZ injections reduced PTZ-kindled convulsions and reversed PTZ-induced pro-oxidative changes in the brain, such as raised levels of malondialdehyde (MDA) or reduced levels of glutathione (GSH). The authors did not mention how many applications of TMZ were given to the mice, but undoubtedly it was more than a single administration [12].

Nevertheless, in a study by Jain et al. [13], TMZ (10 and 20 mg/kg) administered orally only once significantly raised the threshold in the increasing current electroshock seizure (ICES) test in mice. Furthermore, coadministration of TMZ with nimodipine or phenytoin (all drugs applied at subanticonvulsant doses) offered significant antiseizure protection in this test [13]. This encouraged us to test this cytoprotective drug in another model of electrically-induced convulsions—the maximal electroshock test in mice. Moreover, combinations of TMZ with antiepileptic drugs were examined in this test.

In our previous work, we thoroughly studied the interactions between antiepileptic and antiarrhythmic drugs of all classes in the maximal electroshock test in mice, the commonly accepted animal model for studying tonic-clonic convulsions [14]. The involvement of oxidative stress in both cardiovascular disorders and epilepsy [15] might explain the frequent coexistence of the two diseases. For this reason, the common treatment with TMZ and antiseizure medications is very likely. Therefore, we were interested in whether a single administration of TMZ would have any antielectroshock effect and whether it would modify the anticonvulsant action of selected first-generation antiepileptic drugs. Interestingly, the TMZ behaved as a potent antagonist of AMPA/kainate glutamatergic receptors in vestibular ganglion neurons [1]. Such a property may explain the potential antiseizure effects of this cytoprotectant [16].

In this study, we also assessed the effect of TMZ and its combinations with classical antiepileptics on long-term memory and motor coordination in mice. Possible pharmacokinetic interactions were verified by measuring brain concentrations of antiepileptic drugs.

## 2. Results

### 2.1. Electroconvulsive Threshold Test and Maximal Electroshock Test

In the first phase of the study, the influence of TMZ applied at three time points (15 min, 30 min, and 60 min) was tested with the ECT test. Since the greatest antiseizure effect was expressed 15 min after injection with TMZ, this time point was chosen for further examination. The TMZ, applied at a wide dose range, from 20 to 120 mg/kg, did not affect the threshold for electroconvulsions (see Table 1 and Table 2).

In the MES test, TMZ (at doses of 50 mg/kg, 80 mg/kg, 100 mg/kg, and 120 mg/kg) decreased the action of phenobarbital, enhancing its ED_50_ value from 26.9 mg/kg to 38.3 mg/kg, 39.4 mg/kg, 42.5 mg/kg, and 43.6 mg/kg, respectively (F(5.138) = 5.997, *p* < 0.0001). Similarly, this anti-ischemic medication, given at 80 mg/kg, 100 mg/kg, and 120 mg/kg, attenuated the antielectroshock activity of phenytoin, increasing its ED_50_ value from 13.9 mg/kg to 22.3 mg/kg, 26.9 mg/kg, and 27.2 mg/kg, respectively (F(7.152 = 20.018, *p* < 0.0001). TMZ at a dose of 120 mg/kg weakened the effect of carbamazepine, increasing its ED_50_ value from 13.9 mg/kg to 21.7 mg/g (F(6.161) = 6.346, *p* < 0.0001). Only the action of valproate was unaffected by TMZ (see Figure 1A–D).

### 2.2. Chimney Test and Step-Through Passive Avoidance Task

Phenobarbital alone (43.6 mg/kg) and the combination of TMZ (120 mg/kg) with valproate (463.8 mg/kg) significantly impaired motor coordination in mice. TMZ (120 mg/kg) alone, as well as the other antiepileptic drugs and their combinations with TMZ, did not cause any motor deficit (see Table 3).

Concomitant treatment with TMZ (120 mg/kg) and carbamazepine (21.7 mg/kg) resulted in a significant reduction in retention time. However, TMZ (120 mg/kg) alone and in combination with other antiepileptic drugs did not weaken long-term memory in mice. It is worth noting that valproate, carbamazepine, and phenobarbital alone decreased retention time in the passive avoidance task only to an insignificant degree, but the addition of TMZ did not change this trend (see Figure 2).

### 2.3. Brain Concentrations of Antiepileptic Drugs

TMZ (120 mg/kg) did not change the total brain levels of the antiepileptic drugs used in the study (see Figure 3A–D).

## 3. Discussion

The results of the study presented herein indicate that a single administration of TMZ does not affect the electroconvulsive threshold, but, surprisingly, it does decrease the antielectroshock action of carbamazepine, phenytoin, and phenobarbital, without affecting the action of valproate. These revealed interactions seem to be pharmacodynamic, since the TMZ did not change the brain concentrations of any of the antiepileptic drugs used in the study.

In previous investigations, TMZ administered several times exhibited its antiseizure effect against pentylenetetrazole (PTZ)-induced convulsions. Hence, the antioxidant action, the development of which requires some time, can be taken into consideration as a mechanism of the observed effect. Undoubtedly, oxidative stress processes lead to neuroexcitation, neuroinflammation, neurodegeneration, and seizures. Neuroinflammation plays an important role in the process of epileptogenesis through the activation of NMDA (by IL-1β) and AMPA (by TNFα) receptors. Indeed, some antioxidants, like α-tocopherol, ascorbic acid, coenzyme Q10, melatonin, resveratrol, α-lipoic acid, and curcumin have been demonstrated to inhibit the development of some chemically-induced convulsions in experimental animals [17].

However, not only chronic treatment but also single injections of TMZ exhibited some antiseizure action, increasing the threshold of electrically-induced convulsions in the ICES test [13]. Therefore, other “faster” mechanisms, e.g., those that influence receptors or ion channels, are more likely to be involved. As was mentioned before, the antiexcititoxic activity of TMZ was considered to be a mechanism of its protective action in the inner ear [1]. We assumed that a TMZ-induced blockade of AMPA/kainate receptors could contribute to antielectroshock activity, or at least increase the action of the antiepileptic drugs in the MES test. However, nothing of the sort happened. Since AMPA/kainate receptors are widespread in brain areas related to seizures, it may be supposed that TMZ is rather a weak antagonist of these glutamatergic receptors, or that other mechanisms counteract the potential antiseizure action of this cytoprotectant, particularly considering the fact that TMZ even decreased the antielectroshock action of carbamazepine, phenobarbital, and phenytoin. Nevertheless, the possibility that the different actions of TMZ in the ICES and ECT tests are due to methodological differences in the two seizure models cannot be excluded. In the ICES test, every mouse was treated with electrical impulses (20 Hz for 0.2 s), with current intensity linearly increasing from 2 mA in 2 mA/2 s increments, until tonic hindlimb extension occurred. If tonic convulsions were not observed, the test was completed at a current of 30 mA [13]. In the ECT test, three to four groups of animals were subjected to electrical impulses (50 Hz for 0.2 s), with current intensity from 4 mA to 7 mA. One group (eight mice) was treated with a current of the same intensity (for details, see Materials and Methods). From our own experience, the ECT test seems to be more reliable.

Among other mechanisms of action, TMZ has been reported to block dopamine D_2_ receptors [7] and increase the serum and platelet levels of serotonin [18]. Increased brain concentration of serotonin can also modulate seizure processes. This can be analyzed on the example of a representative of selective serotonin uptake inhibitors (SSRIs). A single administration of fluoxetine did not affect the electroconvulsive threshold, but significantly enhanced the anticonvulsive activity of carbamazepine, phenobarbital, and phenytoin, though not that of valproate [19]. Among serotonin and norepinephrine reuptake inhibitors, venlafaxine and milnacipran raised the electroconvulsive threshold, while, if applied at their subprotective doses, the two antidepressants enhanced the antielectroshock action of valproate (venlafaxine) or all four first-generation antiepileptic drugs (milnacipran).

Duloxetine presented typical characteristics of an antiseizure medication at its 50% effective dose (ED_50_) of 48.21 mg/kg [19]. In our opinion, it is unlikely that the anticonvulsant action of all the above-mentioned drugs is due to other mechanisms than serotoninergic ones, for instance, enhancement of GABA-ergic neurotransmission. Therefore, increased serotonin levels in the brain do not seem to be responsible for the attenuation of the antielectroshock action of the antiepileptic drugs used in this study.

In general, dopamine D_2_ receptors have been reported to mediate the antiseizure action of dopamine agonists in different animal epilepsy models, including chemically- and electrically-induced convulsions in mice and rats [20]. However, data on the influence of D_2_ receptor blockades appear somewhat contradictory. In one study, the dopamine D_2_ antagonists remoxipride, raclopride, and haloperidol did not markedly modify seizures induced by PTZ, picrotoxin, or bicuculline [21]. On the other hand, raclopride and haloperidol greatly reduced the threshold for pilocarpine-induced convulsions in another study [22]. Moreover, the selective D2 antagonist raclopride, injected dorsally into both hippocampi, dose-dependently facilitated motor seizures evoked by pilocarpine [23]. Seizures may be precipitated by antipsychotic drugs with D_2_ receptor antagonistic action [24]. It has been observed that a lack of D_2_ receptors alters GABAergic neurotransmission, specifically mediated through GABA_A_ receptors, in the cortex and striatum of mice [25]. Analysis of the above-mentioned data indicates that antagonism towards dopamine D_2_ receptors may be responsible for reducing the antielectroshock effect of carbamazepine, phenobarbital, and phenytoin.

The following question may arise: why does TMZ affect the antielectroshock action of carbamazepine, phenobarbital, and phenytoin, but not that of valproate? This does not appear to result from the underlying mechanisms of action of the antiepileptic drugs. Carbamazepine and phenytoin primarily block voltage-gated sodium channels. Phenobarbital is an agonist of the barbiturate site within the GABA_A_ receptor complex. Valproate combines these two mechanisms by potentiating GABA-ergic neurotransmission and inhibiting intracellular sodium influx [26]. Considering the influence of antiepileptic drugs on oxidative stress, phenytoin, carbamazepine, and phenobarbital exhibit pro-oxidative effects, increasing the brain concentrations of MDA and decreasing those of GSH. On the contrary, valproate showed antioxidant action, enhancing GSH concentrations [27,28,29,30,31]. Nevertheless, the opposite direction of action of TMZ and the three antiepileptic drugs (carbamazepine, phenytoin, and phenobarbital) on redox states cannot explain the decreased antiseizure efficacy of these drugs. In cultured mouse neurons, valproate increased the activity of dopamine D_2_ receptors [32]. Moreover, five-day treatment with valproate was reported to increase the density of dopamine D_2_ receptors in the mouse striatum. According to the authors, this mechanism of action may contribute to the antiseizure action of valproate [33]. In another study, however, 30-day treatment with carbamazepine [34] or valproate [35] was reported to downregulate D_2_ receptors in the rat brain. Nevertheless, it appears that the reduced antielectroshock action of first-generation antiepileptic drugs may be related to dopaminergic mechanisms. Further extensive investigation is required to confirm this assumption. It will be also necessary to evaluate the effect of chronically administered TMZ on the action of antiepileptic drugs in the MES test. Hopefully, this will clarify whether and how the antioxidant action of TMZ, not developed after a single administration, modifies the interactions observed in the present study.

Interestingly, valproate was the only antiepileptic drug in this study whose combination with TMZ led to a significant motor deficit. It should be emphasized, however, that valproate alone had an insignificant tendency to impair motor coordination. From the theoretical point of view, this tendency is most likely related to valproate-induced enhancement of the central GABA-ergic neurotransmission [36]. On the other hand, it has been reported that inducible loss of D_2_ receptors causes significant motor impairment in mice [37,38]. Bearing in mind the fact that TMZ blocks dopaminergic D_2_ receptors [7], one can assume that the addition of the two mechanisms can trigger motor impairment in animals treated concomitantly with valproate and TMZ. Nevertheless, this hypothesis does not seem to be reliable in light of results for phenobarbital, a potent agonist of the barbiturate site within the GABA-related complex. This antiepileptic, applied alone, slightly impaired motor coordination in mice, but its combination with TMZ did not produce significant motor deficits. Therefore, other mechanisms, unknown at this stage of our knowledge, must be responsible for the severe motor disorders induced by the combination of valproate and TMZ.

In conclusion, as far as the obtained results can be transferred to clinical conditions, TMZ does not seem to be a beneficial anti-ischemic drug for patients with IHD and epilepsy, even though the combined treatment of TMZ and antiepileptic drugs did not cause any significant undesired effects. This finding appears to be quite disappointing, since drugs with cytoprotective properties would be very desirable in the treatment of epilepsy, which is considered one of the most common neurodegenerative diseases. However, it should be remembered that drug doses used in experimental studies are usually higher than the recommended doses for clinical use. After conversion to the human dose [39], the lowest dose of TMZ effective in the MES test (50 mg/kg) remains 4 times higher than the daily clinical dose for this drug (70 mg).

## 4. Materials and Methods

### 4.1. Animals

The experiments in this study were conducted on female Swiss mice weighing 20–25 g and approved for testing by a veterinarian. The rodents were housed in colony cages in which they were provided with appropriate laboratory conditions. They were housed in colony cages, had a natural day-night cycle of 12/12 h, and constant access to water and food. Temperature oscillated in the range of 20–24 °C, while air humidity was 45–65%. Air was exchanged 15 times/h. The required acclimatization time was 7 days. Then, the mice were randomly matched to form groups (8–10 animals per group). The methodology of the experiments used in this study was approved by the Local Ethical Committee at the University of Life Sciences in Lublin (license No 67/2016) and met the requirements of ARRIVE guidelines as well as EU Directive 2010/63/EU for animal experiments.

### 4.2. Drugs

Trimetazidine (TMZ), valproate (VPA), carbamazepine (CBZ), phenytoin (PHT), and phenobarbital (PB) were used in the present study. VPA, CBZ, and PHT were obtained from Sigma (St. Louis, MO, USA), PB from UNIA Pharmaceutical Department (Warsaw, Poland), and TMZ from ANPHARM Pharmaceutical Joint-Stock Company (Warsaw, Poland). All medications were suspended in a 1% aqueous solution of Tween 80 (Sigma-Aldrich, St. Louis, MO, USA). Then, drugs or vehicles (in control groups) were administered intraperitoneally (ip) in a volume of 10 mL/kg body weight. TMZ was administered 15 min prior to tests. Among the antiepileptic drugs, VPA and CBZ were administered 30 min, PB 60 min, and PHT 120 min before experiments.

Drug doses and their application times were based on our experimental experiences published elsewhere [40], whereas those for the TMZ were initially taken from the literature, but ultimately experimentally specified.

### 4.3. Maximal Electroshock Seizure Test

The test of maximal electroshock seizures (MES) is categorized as animal model of generalized tonic-clonic seizures. Its use is obligatory in initial preclinical testing of potential anticonvulsant drugs [14].

In the first stage of the study, the effect of TMZ on electrical convulsions was evaluated in the electroconvulsive threshold test (ICT). The threshold for electroconvulsions was expressed as CS_50_, which is a current strength (mA) necessary to induce tonic convulsions in 50% of animals. To determine this parameter, at least four groups of mice were subjected to currents of intensities in the range of 4–7 mA (the same intensity value for one group of animals). Subsequently, an intensity–response curve was calculated on the basis of the percentage of convulsing animals.

The anticonvulsant properties of the antiepileptic drugs and their combinations with TMZ were assessed in the MES test and expressed as ED_50_, a median effective dose saving 50% of animals from tonic convulsions. To determine respective ED_50_ values, no less than three groups of mice were treated with increasing doses of antiepileptic drugs given alone or in conjunction with TMZ. Then, the mice were subjected to a current of 25 mA and a dose–effect curve was determined based on the percentage of mice that avoided seizures (protective effect below, around, and above 50%), according to Litchfield and Wilcoxon [41].

The other current parameters common to the ECT and MES were: frequency of 50 Hz, voltage of 500 V, and duration of 0.2 s. Electrical impulses were produced by a licensed rodent shocker (Hugo Sachs Elektronik, Freiburg, Germany) and delivered to the animals by ear electrodes.

### 4.4. Chimney Test

Motor coordination in the mice was determined in the chimney test [42]. On the first day of the experiment (adaptation day), untreated mice were put separately in a horizontally positioned plastic cylinder with internal threading, an inner diameter of 3 cm, and a length of 25 cm. When the animals came to the end of the pipe, it was positioned vertically, hence the mouse had to climb backwards to get out of the chimney. Rodents that failed to do so within 60 s were excluded from further experiments. On the second day, the same test was carried out after drug administration. Rodents were administered TMZ, antiepileptic drugs (providing 50% protection against convulsions), and antiepileptics combined with TMZ (still ensuring 50% protection). This made it possible to determine whether TMZ impairs motor coordination or modifies the influence of antiepileptic drugs on this parameter. Results of the chimney test are presented as a percentage of mice that did not accomplish the test.

### 4.5. Step-Through Passive-Avoidance Test

This test is used to evaluate long-term memory in rodents, utilizing their natural preference for dark places [43]. The mice were examined in a computer-controlled apparatus (Multi Conditioning System, MCS, TSE Systems GmbH, Bad-Homburg, Germany) that electronically counted the time of passage of the mouse into a dark room. The equipment ensured the complete isolation of the rodents from external distractors (olfactory, auditory, and visual stimuli), including the presence of a researcher. This in turn increased the reliability of the results.

The MCS apparatus contains two chambers, illuminated and darkened, connected by a sliding door. On the first day of the procedure, mice were administered TMZ, antiepileptic drugs, and combinations of antiepileptic drugs and TMZ, as in the chimney test. Then, the animals were placed separately in the light compartment and allowed to move freely. After complete transition to the dark room, the movable door was closed, and a mild electrical impulse (0.3 mA) was conducted for 2 s through the bars in the floor. Immediately after that procedure, each mouse was removed from the apparatus and inserted into the home cage. On the following day (24 h later), pre-tested animals were again placed in the light chamber, but the door was open during the whole experiment. Transition to the dark compartment within 180 s was automatically registered and treated as a sign of long-term memory deficit. The results of the passive-avoidance task were expressed as medians (with 25th and 75th percentiles) of the time spent in the bright room.

### 4.6. Measurement of Brain Concentrations of Antiepileptic Drugs

To verify possible pharmacokinetic interactions between the TMZ and the antiepileptic drugs, total concentrations of the latter drugs were assessed in the mouse brains. Animals from the control groups were administered a respective antiepileptic drug and vehicle. The test groups were given TMZ (120 mg/kg) instead of the vehicle. At the time of the maximum effect of the antiepileptic drugs, the mice were decapitated, and their brains were then extracted from their skulls. Subsequently, the brains were homogenized (Ultra Turax T8 homogenizer, IKA, Staufen, Germany) with Abbott buffer (2:1 vol/weight) and centrifuged at 10,000× *g* for 10 min. The supernatants obtained (75 µL) were used for further analysis by fluorescence polarization immunoassay (Architect c4000 analyzer, Abbott Laboratories, Poland). Antiepileptic drug concentrations were recorded as means ± SD (µg/mL) of eight determinations.

### 4.7. Statistics

The values of the ED_50_s with 95% confidence limits were calculated in the computer log-probit evaluation based on the method of Litchfield and Wilcoxon [41]. Then, the confidence limits were converted to standard errors (SEMs), and subsequent multiple comparisons of the ED_50_ values (± SEM) were carried out using the one-way analysis of variance (ANOVA) followed by a post hoc Tukey test. Regarding the results obtained in the chimney test and passive avoidance task, we used, respectively, Fisher’s exact probability test for qualitative variables and a non-parametric Kruskal–Wallis test. Statistical analysis of changes in brain concentrations of antiepileptic drugs was conducted using an unpaired Student’s *t* test. The results were considered significant if *p* was at least less than 0.05.

## Figures and Tables

**Figure 1 ijms-23-11328-f001:**
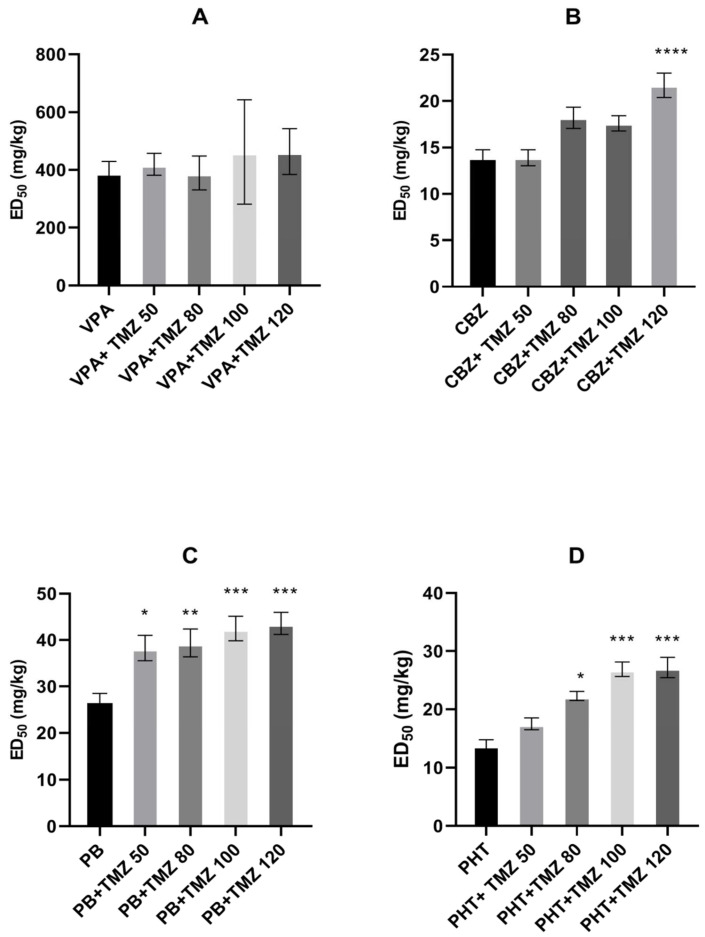
Effect of trimetazidine (TMZ) on the anticonvulsant action of (**A**) valproate (VPA), (**B**) carbamazepine (CBZ), (**C**) phenobarbital (PB), and (**D**) phenytoin (PHT) against maximal electroshock-induced seizures in mice. Data are presented as median effective doses (ED_50_ in mg/kg) with standard errors (SEM) that protect 50% of animals from the tonic-clonic seizures. Statistical analysis of the data was performed using one-way analysis of variance (ANOVA) followed by a post hoc Tukey test. * *p* < 0.05, ** *p* < 0.01, *** *p* < 0.001, **** *p* < 0.0001 vs. control group.

**Figure 2 ijms-23-11328-f002:**
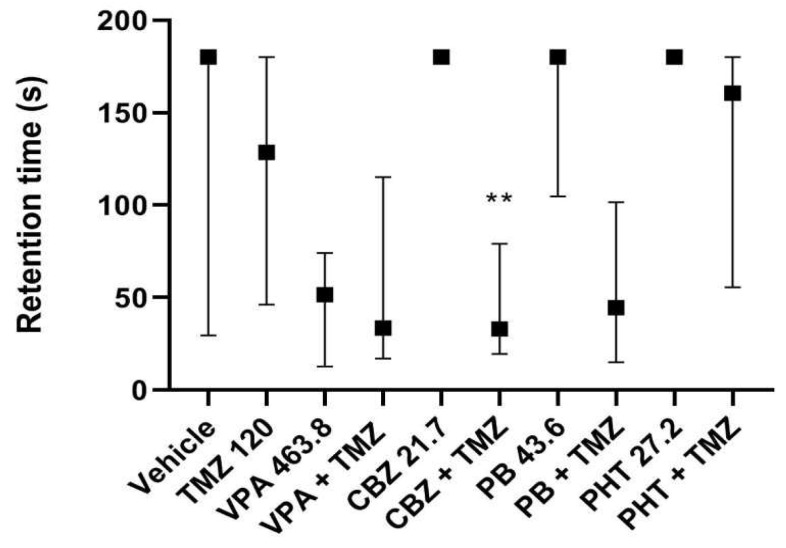
Results are shown as median retention time (with 25th and 75th percentiles) during which mice avoided the dark room in the step-through passive-avoidance task. Drug doses used in this test reflect ED50 doses of combinations of TMZ (120 mg/kg) with the antiepileptic drugs from the MES test. Statistical analysis of data was performed using a nonparametric Kruskal–Wallis ANOVA test followed by Dunn’s post hoc test. TMZ—trimetazidine, VPA—valproate, CBZ—carbamazepine, PB—phenobarbital, PHT—phenytoin, ** *p* < 0.01 vs. CBZ.

**Figure 3 ijms-23-11328-f003:**
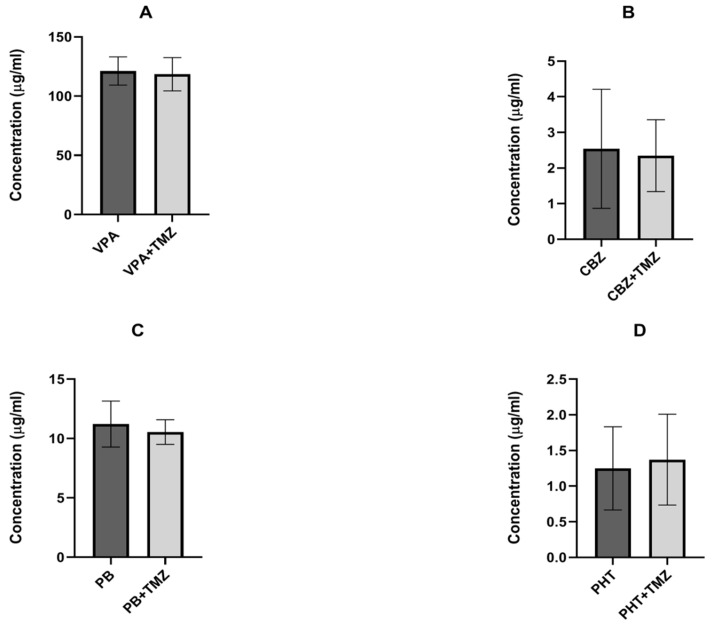
Effect of trimetazidine (TMZ) on the brain levels of (**A**) valproate (VPA), (**B**) carbamazepine (CBZ), (**C**) phenobarbital (PB), and (**D**) phenytoin (PHT). Results are presented as the ±SD (in µg/mL) of at least eight determinations. Statistical analysis was performed using an unpaired Student’s test. TMZ—trimetazidine, VPA—valproate, CBZ—carbamazepine, PB—phenobarbital, and PHT—phenytoin.

**Table 1 ijms-23-11328-t001:** Effect of trimetazidine (40 mg/kg), applied at different time points, on the electroconvulsive threshold in mice.

Drug (mg/kg)	CS_50_ ± SEM (mA)	Detailed Data
Vehicle	5.8 ± 0.42	5 mA 1/8 6 mA 4/8 7 mA 7/8
TMZ 60 min	5.9 ± 0.42	5 mA 2/8 6 mA 4/8 7 mA 6/8
TMZ 30 min	5.8 ± 0.37	5 mA 2/8 6 mA 4/8 7 mA 7/8
TMZ 15 min	5.2 ± 0.34	4 mA 1/8 5 mA 3/8 6 mA 4/8 7 mA 8/8

Results are presented as median current strength (CS_50_ with SEM) producing tonic convulsions in 50% of animals. Detailed data reflect the number of animals with tonic convulsions (per eight animals in the group) using a given current intensity. TMZ, trimetazidine, was injected ip 60 min, 30 min, and 15 min before the test.

**Table 2 ijms-23-11328-t002:** Effect of trimetazidine on the electroconvulsive threshold in mice.

Drug (mg/kg)	CS_50_ ± SEM (mA)
Vehicle	5.4 ± 0.34
TMZ (20)	5.4 ± 0.25
TMZ (40)	5.1 ± 0.40
TMZ (80)	5.2 ± 0.41
TMZ (120)	5.4 ± 0.25

Results are presented as median current strength (CS_50_ with SEM) producing tonic convulsions in 50% of animals. TMZ, trimetazidine, was injected ip 15 min before the test.

**Table 3 ijms-23-11328-t003:** Effect of trimetazidine and its combinations with classical antiepileptic drugs on retention time in the passive avoidance task and motor impairment in the chimney test in mice.

Drugs (mg/kg)	Mice Impaired (%)
Vehicle	0
TMZ (120)	0
VPA (463.8)	40
VPA (463.8) + TMZ (120)	100 ***^,#^
CBZ (21.7)	40
CBZ (21.7) + TMZ (120)	20
PB (43.6)	50 *
PB (43.6) + TMZ (120)	40
PHT (27.2)	30
PHT (27.2) + TMZ (120)	10

Results are presented as percentage of animals that failed to perform the chimney test. Statistical analysis of data was performed using Fisher’s exact probability test. Drug doses used in the chimney test reflect ED_50_ doses of combinations of TMZ (120 mg/kg) with the antiepileptic drugs from the MES test. TMZ—trimetazidine, VPA—valproate, CBZ—carbamazepine, PB—phenobarbital, PHT—phenytoin, ^#^
*p* < 0.05 vs. VPA, * *p* < 0.05 vs. vehicle and TMZ; *** *p* < 0.001 vs. vehicle and TMZ.

## Data Availability

The data presented in this study are available in the article.

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
