# Peer review of "Trimetazidine, an Anti-Ischemic Drug, Reduces the Antielectroshock Effects of Certain First-Generation Antiepileptic Drugs"

_ijms, 2022, doi:10.3390/ijms231911328_

Round 1
Reviewer 1 Report
The manuscript titled “Trimetazidine, an anti-ischemic drug, reduces the antielectro-2 shock effects of certain first-generation antiepileptic drugs” by Kinga Borowicz-Reutt and Monika Banach is well written and focused on some interesting aspects of combinatorial approach for the treatment of seizures. However, the study reached the conclusion that the combination approach of TMZ with other antiepileptic drugs may not work as an add therapy.
I have some concerns regarding the data representation and conclusions dependent on that data; and some minor comments.
1. The data given in Table 2 (retention time and % mice impaired) is difficult to visualize and interpret in the tabular form. It would be better if the data is in graphical form, especially with individual data points. Also, looking at the values of retention time, it is surprising that there was no statistical difference among the groups.
2. In the same Table 2, why the data is expressed as ‘median’ values, instead of ‘mean’? Most of the passive avoidance studies’ data are expressed as mean. Is there any specific reason to do so?
3. In the chimney test for motor impairment, TMZ caused further motor impairment when combined with VPA unlike with CBZ, PB or PHT. The authors did not clearly explain the reason for such a discrepancy.
4. In the results (line #91), the authors mention that “the antiseizure effect was most expressed 15 min after injection with TMZ, this time point was chosen for further examinations”. However, the relevant data is not provided. If it has been concluded based on a previous study, please provide the reference.
5. Line #170: “Single administration of fluoxetine …………but not that of valproate”. The reference is not provided for the statement.
6. Texts in the methods section, especially methods 4.3 and 4.5, are exact copies of the previous paper of the authors. The authors may want to change the flow of text.
Author Response
I would like to express my thanks for all valuable comments ant time spent on review.
Referring to individual comments:
- The data given in Table 2 (retention time and % mice impaired) is difficult to visualize and interpret in the tabular form. It would be better if the data is in graphical form, especially with individual data points. Also, looking at the values of retention time, it is surprising that there was no statistical difference among the groups.
Results from the passive-avoidance task have been removed from the Table 2 (now being the Table 3) and presented in the Figure 3. We have done it similarly to a graph presenting data from our lab (Luszczki et al., Neuropsychopharmacology. 2003 Oct;28(10):1817-30). A graph with individual data points seems to be unreadable… But, of course, we will swap figures, if it is more advantageous.
Indeed, visual comparison of retention times can be confusing, probably because the results are presented as medians with percentiles. Nevertheless, statistical analysis of the results has been properly carried out. Unfortunately, one error appeared in the numerical data. The median retention time for CBZ + TMZ 120 should be 33.3 instead of 53.3. Of course, this mistake has been corrected in the graphical form.
- In the same Table 2, why the data is expressed as ‘median’ values, instead of ‘mean’? Most of the passive avoidance studies’ data are expressed as mean. Is there any specific reason to do so?
To the best of our knowledge, descriptive statistics such as means and standard errors can only be provided if the described data are drawn from a general population with normal distribution. Otherwise median values with quartiles should be provided. We checked the normality of data distribution with Shapiro-Wilks test and decided to use non-parametric test for further statistical analysis. Non-parametric methods have a great advantage in that they can be applied irrespective of the distribution form (Hamada, J Toxicol Pathol 2018, 31: 15–22; CichoÅ„, Pharmacological Reports 2020, 72:481–485).
In our lab, we have been using medians with percentiles for years. Consistent use of one method made it possible to compare all the obtained data which in time became an additional benefit.
- In the chimney test for motor impairment, TMZ caused further motor impairment when combined with VPA unlike with CBZ, PB or PHT. The authors did not clearly explain the reason for such a discrepancy.
I tried my best to discuss this phenomenon in the penultimate paragraph of the discussion section. Nevertheless I am aware of all weak points of this attempt.
Any further suggestions from the reviewer are welcome.
- In the results (line #91), the authors mention that “the antiseizure effect was most expressed 15 min after injection with TMZ, this time point was chosen for further examinations”. However, the relevant data is not provided. If it has been concluded based on a previous study, please provide the reference.
That data has been provided in Table 1.
- Line #170: “Single administration of fluoxetine …………but not that of valproate”. The reference is not provided for the statement.
The appropriate reference was provided.
- Texts in the methods section, especially methods 4.3 and 4.5, are exact copies of the previous paper of the authors. The authors may want to change the flow of text.
The text in the methods was changed. Thank you for this comment. Sometimes it is not easy to express the same things in other words.
The manuscript has been corrected by a native English-speaking person (professional English language translator).
Reviewer 2 Report
The study aims to assess the influence of TMZ concomitant use with four antiepileptic drugs on long-term memory and motor coordination in mice. The topic of this manuscript presents high scientific value and originality, but some of the areas need to be improved.
1. Compiling the chimney test and PA task results into one table is confusing, especially when the Materials and Methods section of the manuscript presents those approaches separately. I propose to introduce an additional table or figure to resolve this issue.
2. Abbreviation ICES (current electroshock seizure) which is used two times (one in the Introduction and one in the Discussion section), should be used with the link to the results it refers to and also present in the Materials and methods section if appropriate. Now it is unclear if it’s a result of the study or only a reference. It needs to be clarified what MES and ICES are.
3. In the MES test, the mice received TMZ in the 20, 40, 80 and 120 mg/kg doses. Why are doses presented in figure 1 a-d related to ED50, different? In this case, the authors used 50, 80, 100, and 120 mg/kg, so there is a difference in the case of 20 and 50 mg. It needs to be explained.
4. Why, in the Chimney test, PS task and brain concentration analysis, was only 120 mg TMZ used?
5. Drug doses used to analyse the retention time and mice impaired need to be explained in the text or in figure 1, which is a better solution.
Author Response
I would like to express my gratitude for the review and all the valuable remarks.
Referring to comments:
- Compiling the chimney test and PA task results into one table is confusing, especially when the Materials and Methods section of the manuscript presents those approaches separately. I propose to introduce an additional table or figure to resolve this issue.
Thank you for this comment. Results from the passive-avoidance task were removed from the table and were presented in a form of graph.
- Abbreviation ICES (current electroshock seizure) which is used two times (one in the Introduction and one in the Discussion section), should be used with the link to the results it refers toand also present in the Materials and methods section if appropriate. Now it is unclear if it’s a result of the study or only a reference. It needs to be clarified what MES and ICES are.
Results obtained from the ICES test were described by Jain et al. (2010). The respective reference has been inserted in the end of this paragraph. Some more information about ICES and ECT and MES tests has been added, particularly in the Discussion section. Also, the respective subsection of Materials and Methods has been expanded.
- In the MES test, the mice received TMZ in the 20, 40, 80 and 120 mg/kg doses. Why are doses presented in figure 1 a-d related to ED50, different? In this case, the authors used 50, 80, 100, and 120 mg/kg, so there is a difference in the case of 20 and 50 mg. It needs to be explained.
TMZ was used at doses of 20, 40, 80, and 120 mg/kg in the electroconvulsive threshold test. Doses presented in Figure 1 refer to the MES test. Differences between the two tests, or rather two phases of the same test, have been described in more detail in Materials and Methods. Doses of 20, 50, 80, and 120 mg/kg were used in the MES test.
In the beginning of this study, we knew little about the dosage of TMZ in behavioral tests. Therefore, we started from doses known from literature – 20 and 40 mg/kg. Subsequently, doses were adjusted to our needs. In further experiments (the MES test), we focused on achieving the dose-effect relationship by using a minimum number of animals.
- Why, in the Chimney test, PS task and brain concentration analysis, was only 120 mg TMZ used?
In the abovementioned tests, we mostly apply a tested drug at its highest dose used in combinations with antiepileptic drugs. Thus, probability of manifestation of undesired effects and pharmacokinetic interactions is the highest. Simultaneously, this allows for maximum reduction in animal use.
- Drug doses used to analyse the retention time and mice impaired need to be explained in the text or in figure 1, which is a better solution.
It is also our common practice. Drug doses used in the chimney and passive-avoidance reflect ED50 doses of combinations of TMZ (120 mg/kg) with antiepileptic drugs obtained from the MES test. Thus, we can evaluate undesired effects caused by combinations providing 50% protection against MES-induced seizures. As controls we use antiepileptic drugs alone applied at doses used in such combinations.
Appropriate explanations were added to legends to Table 3 and Figure 2.
The manuscript has been corrected by a native English-speaking person (professional English language translator).
Reviewer 3 Report
General comment:
Like in many experimental studies, the applied dose of TMZ was much higher than the drug recommended dose for clinical use. Therefore, extrapolation of the observed effects to clinical situation is affected by bias of the dose.
Major concern:
Apart of the doses applied in the current study, only one model was applied. However, antiepileptic drugs that protect against partial and nonconvulsive seizures in epileptic patients failed to do so in the maximal ES (MES) model. Therefore, to provide more conclusive observations other models should be applied.
The authors should possess the brain tissues form the animals, so that some potential mechanisms debated in the discussion can be directly addressed (why different interaction of TMZ was observed with the studied drugs, i.e. valproate against other studied), making the paper more suitable for IJ of Molecular Sciences.
Minor:
English should be polished, examples from the abstract „ did not impair the motor impairment” or “The obtain results suggest ..”
Author Response
At first, I would like to express my thanks for all valuable comments ant time spent on review.
Referring to individual comments:
General comment:
Like in many experimental studies, the applied dose of TMZ was much higher than the drug recommended dose for clinical use. Therefore, extrapolation of the observed effects to clinical situation is affected by bias of the dose.
I highly agree with this comment. The appropriate sentence has been inserted in the end of the Discussion section.
Major concern:
Apart of the doses applied in the current study, only one model was applied. However, antiepileptic drugs that protect against partial and nonconvulsive seizures in epileptic patients failed to do so in the maximal ES (MES) model. Therefore, to provide more conclusive observations other models should be applied.
Thank you for this comment. Actually, we plan to extend this study to the pentylenetetrazole-kindled seizures in mice. Not only acute, but also chronic administration of TMZ is taken into consideration.
The authors should possess the brain tissues form the animals, so that some potential mechanisms debated in the discussion can be directly addressed (why different interaction of TMZ was observed with the studied drugs, i.e. valproate against other studied), making the paper more suitable for IJ of Molecular Sciences..
Actually, we did it. We isolated the hippocampus, frontal cortex, hypothalamus and the heart tissue for genetic testing. Such studies are, however, time- and cost-consuming. Hopefully, results will be published in the next article.
Minor:
English should be polished, examples from the abstract „ did not impair the motor impairment” or “The obtain results suggest ..”
The manuscript has been corrected by a native English-speaking person (professional English language translator).
Round 2
Reviewer 2 Report
I am satisfied with the introduced changes. However, I would like to suggest Authors move all of the dosage explanations to the Material and Methods, 4.2 Drugs section instead of providing scattered descriptions throughout the manuscript. The paper will gain clarity on the findings.
Author Response
Yes, such scattered information about drug doses could be confusing. All explanations from the 2.1 Result section and legends to Table 3 and Figure 2 have been moved to the 4.2 Drugs section.
Thank you very much for this suggestion.
Reviewer 3 Report
As the revision time was short, the authors were able to improve presentation of the results and implement English editing. However, the main concerns of the submision do remain. Only one model is evaluated (but the authors plan to study chronic TMZ adminsitration and in other models) or the tissues are at the authors tissue bank (but the authors plan to analyse the samples in the future). I would propose to offer the authors longer time for resubmission, and publish higher quality manuscript which would enable more consistent conclusions.